# Osteometabolism: Metabolic Alterations in Bone Pathologies

**DOI:** 10.3390/cells11233943

**Published:** 2022-12-06

**Authors:** Rupesh K. Srivastava, Leena Sapra, Pradyumna K. Mishra

**Affiliations:** 1Translational Immunology, Osteoimmunology & Immunoporosis Lab (TIOIL), Department of Biotechnology, All India Institute of Medical Sciences (AIIMS), New Delhi 110029, India; 2Department of Molecular Biology, ICMR-NIREH, Bhopal 462030, India

**Keywords:** Osteometabolism, bone cells, metabolism, bone pathologies, gut-associated metabolites (GAMs)

## Abstract

Renewing interest in the study of intermediate metabolism and cellular bioenergetics is brought on by the global increase in the prevalence of metabolic illnesses. Understanding of the mechanisms that integrate energy metabolism in the entire organism has significantly improved with the application of contemporary biochemical tools for quantifying the fuel substrate metabolism with cutting-edge mouse genetic procedures. Several unexpected findings in genetically altered mice have prompted research into the direction of intermediate metabolism of skeletal cells. These findings point to the possibility of novel endocrine connections through which bone cells can convey their energy status to other metabolic control centers. Understanding the expanded function of skeleton system has in turn inspired new lines of research aimed at characterizing the energy needs and bioenergetic characteristics of these bone cells. Bone-forming osteoblast and bone-resorbing osteoclast cells require a constant and large supply of energy substrates such as glucose, fatty acids, glutamine, etc., for their differentiation and functional activity. According to latest research, important developmental signaling pathways in bone cells are connected to bioenergetic programs, which may accommodate variations in energy requirements during their life cycle. The present review article provides a unique perspective of the past and present research in the metabolic characteristics of bone cells along with mechanisms governing energy substrate utilization and bioenergetics. In addition, we discussed the therapeutic inventions which are currently being utilized for the treatment and management of bone-related diseases such as osteoporosis, rheumatoid arthritis (RA), osteogenesis imperfecta (OIM), etc., by modulating the energetics of bone cells. We further emphasized on the role of GUT-associated metabolites (GAMs) such as short-chain fatty acids (SCFAs), medium-chain fatty acids (MCFAs), indole derivates, bile acids, etc., in regulating the energetics of bone cells and their plausible role in maintaining bone health. Emphasis is importantly placed on highlighting knowledge gaps in this novel field of skeletal biology, i.e., “Osteometabolism” (proposed by our group) that need to be further explored to characterize the physiological importance of skeletal cell bioenergetics in the context of human health and bone related metabolic diseases.

## 1. Introduction

Bone is a dynamic organ that undergoes continual remodeling to sustain the homeostatic balance between bone-forming osteoblast and bone-resorbing osteoclast cells. Through the synthesis of osteoid molecules and the release of matrix vesicles, osteoblasts encourage mineralization and bone production. On the other hand, bone-resorbing osteoclast cells have the ability to release organic acids and proteases to digest and resorb the bone matrix. The most abundant cells in bone tissue, called osteocytes, are crucial for bone matrix renewal and maintenance. Osteocytes produce various signals that induce osteoblastogenesis such as prostaglandin E2 (PGE2), growth factors, glycoproteins, etc. [1]. In contrast, osteocytes also negatively regulate osteoblasts by producing sclerostin protein which acts on the osteoblasts and inhibits bone formation. Sclerostin inhibits the Wnt-β-catenin signaling pathway by binding to the low-density lipoprotein receptor-related protein (LRP) 5/6 receptor present on the osteoblast membrane [2]. Moreover, osteocytes also secrete receptor activators of nuclear factor B ligand (RANKL) which induces osteoclastogenesis. Thus, proper functioning of the skeleton system requires the critical management of bone remodeling as irregularities in the bone remodeling process result in various skeleton deformities. Several biochemical factors such as parathyroid hormone (PTH), estrogen, glucocorticoids, insulin growth factor (IGF), bone morphogenetic protein (BMP), transforming growth factor (TGF)-β, and cytokines are involved in the regulation of bone remodeling [3,4]. Recent advancements demonstrated that in addition to the respective signaling pathways and enhanced expression of genes in these bone cells, differentiation of osteoblasts and osteoclasts further depends on metabolic reprogramming. Production of energy from the metabolic pathways such as glycolysis, oxidative phosphorylation (OXPHOS), fatty acid oxidation, glutamine metabolism, etc., not only supports the phenotypic alterations observed during differentiation from osteoprogenitors to mature bone cells but also regulates the functional activity of these cells. Recently, a study reported that under mechanical stimulatory conditions, osteocytes secrete citrate that promotes the activation of osteoblasts, whereas a high glucose environment impaires the effect of osteocytes [5]. Emerging studies categorized bone-related diseases such as osteoporosis, rheumatoid arthritis (RA), spondylitis, etc., into metabolic diseases. To understand how the metabolic alterations are associated with the bone turnover markers such as cross-linked C-telopeptide of type I collagen (CTX) and procollagen type I N- propeptide (P1NP), high-resolution metabolomics approaches are followed. Upon characterization, it has been observed that metabolites linked with the P1NP were associated with metabolic pathways such as tricarboxylic acid (TCA), pyruvate metabolism, vitamin B metabolism, etc. [6]. Whereas metabolites correlated with the CTX levels were enriched with lipid and fatty acid β-oxidation. In addition, CTX levels were linked with the vitamin D and E metabolism and with the biosynthesis of bile acids (BA) [6], thereby highlighting the importance of metabolic alterations in the population. As we continue to assimilate a greater understanding of the complex mechanisms governing bone metabolism and the key regulators involved, we may also gain insights into the causes of many bone-related illnesses. Osteo-metabolism is proposed here as a novel field that highlights the importance of metabolism in bone cells, especially in the context of how dysregulation of metabolic processes leads to the development of various metabolic bone loss pathologies. In the following sections, we discuss the metabolic requirements of bone cells and how alterations in their metabolic pathways govern the progression and development of bone-related diseases.

## 2. Osteometabolism: Metabolic Requirements of Bone Cells

Emerging shreds of evidence revealed that during differentiation, osteoblasts retrieve energy from glycolysis (glucose metabolism) and oxidative phosphorylation (OXPHOS-mitochondrial respiration), whereas mature osteoblasts obtain energy only from glycolysis to synthesize collagen and matrix mineralization. During osteoblast differentiation, the Wnt signaling pathway alters the cellular metabolism by inducing aerobic glycolysis, fatty acid β-oxidation, and glutamine catabolism in osteoblast lineage cells [7]. Various evidence suggests that the peroxisome proliferator-activated receptor gamma (PPARγ) plays a crucial role in skeletal homeostasis. PPARγ inhibits osteoblast differentiation from mesenchymal stem cells (MSCs) [8]. In addition, a study reported that PPARγ increases the differentiation of osteoclasts from the hematopoietic stem cells (HSCs) [9]. A study reported that the loss of PPARγ function in mouse hematopoietic lineages leads to impaired osteoclasts differentiation, resulting in the development of osteopetrosis; a disease exemplified by enhanced bone mass [10]. In contrast to this, the gain of PPARγ function by rosiglitazone (BRL) (PPARγ-ligand) enhanced osteoclastogenesis and bone resorption under both in vitro and in vivo conditions. In 2010, Wei et al., reported that PPARγ performs tissue-specific functions that are further regulated by the coactivator peroxisome proliferator-activated receptor-gamma coactivator 1β (PGC-1β) [10]. PGC-1β is a transcriptional coactivator that regulates energy metabolism via stimulating mitochondrial biogenesis and thus stimulates osteoclast differentiation [10,11].

Osteoclasts have a great degree of mobility and their differentiation from osteoclast precursors and bone resorption are found to be energy-intensive processes that calls for an active metabolic reprogramming. These factors have a well-established role in mitochondrial biogenesis, and it has been observed that deficiency of these factors impairs osteoclast differentiation and bone resorption machinery [10]. In response to RANKL, PPARγ is known to promote osteoclast development and activity via increasing the expression of c-fos and the nuclear factor of activated T cells 1 (NFATc1) signaling (Figure 1). Though mitochondria generate energy from the OXPHOS for osteoclasts differentiation, a study demonstrated that inhibition of mitochondrial complex I with rotenone enhanced its functional activity. This clearly suggests the existence of a metabolic switch between glycolysis and OXPHOS that fine-tunes the differentiation and functional transitions of osteoclasts. Altogether, these studies highlight the importance of energy metabolism in the differentiation and functional activity of bone cells. In the following sections, we discuss the metabolic characteristics of bone cells and the mechanisms governing the usage of energy substrates such as glucose, fatty acids, glutamine, etc.

### 2.1. Glucose Metabolism in Bone Cells

Recently, it has been discovered that, after the liver, skeletal muscle, and fat, the skeleton system is the fourth-largest organ to consume glucose. Glucose serves as a predominant source for the development of bone and therefore plays a vital role in skeleton homeostasis. After being transported into the cells via the glucose transporters (Glut), glucose is metabolized in the cytoplasm by glycolysis to obtain two pyruvate molecules, two adenosine triphosphate (ATP), and two reducing equivalents in the form of nicotinamide adenine dinucleotide (NADH). In addition, the glycolytic pathway produces several intermediates which are crucial for the multiple metabolic pathways and supports the physiological functions of the host. During glycolysis, glucose is metabolized by a sequential series of reactions that produce ATP. Pyruvate, a by-product of glycolysis, travels through the TCA cycle and is transformed in the mitochondria into acetyl-CoA under aerobic conditions and into lactate under anaerobic conditions.

A growing body of evidence points to glycolysis as the primary metabolic process responsible for supplying the significant amount of ATP needed for osteoblast development [12,13]. Glut-1 and Glut-3 are the main glucose transporters that promote the absorption of glucose in osteoblasts lineage cells. Uptake of glucose via Glut is an energy-independent process that transports glucose along a concentration gradient. Recently, Glut-1 was shown to be involved in preventing the AMP-activated protein kinase (AMPK) mediated proteasomal degradation of Runx2 (osteoblast differentiation transcription factor) and demonstrated that the deletion of Glut1 in the osteoblast precursor suppressed its differentiation into mature osteoblasts under both in vitro and in vivo conditions [14] (Figure 2). Additionally, it has been noted that Runx2 is unable to stimulate osteoblast development in the absence of glucose. However, the mechanism underlying Glut-3-mediated glucose absorption and osteoblast differentiation has not yet been studied. In 2017, Lee et al. reported that Wnt signaling promotes glycolysis in osteoblasts via the mammalian target of the rapamycin (mTOR) signaling pathway, whereas parathyroid hormone (PTH) promotes glycolysis via the insulin-like growth factor-1 (IGF-1) mediated signaling pathway. Surprisingly, historical studies have shown that lactate is the final product of glucose oxidation irrespective of the presence or absence of oxygen. According to a recent study, aerobic glycolysis provides 80% of the energy required by osteoblasts, and malic enzyme 2 (Me2) directs glucose carbons to the malate aspartate shuttle to maintain glycolysis [15]. More recently, a study showed that limiting carbohydrates for a short period decreased the levels of circulating bone growth indicators during the resting phase [16]. Together, these studies highlight the importance of glucose metabolism in regulating the differentiation and functional activity of osteoblasts.

It has been reported that the fusion of monocyte precursors to generate osteoclasts is an energy-dependent process that is supported by the enhancement of the number and size of mitochondria. During osteoclastogenesis, the expression of Glut-1 and glutamine transporters was observed to be enhanced with a subsequent increase in the genes responsible for glycolysis and glutaminolysis [17]. In addition, a study showed that increased glycolysis and lactate generation are only essential for bone resorption [18]. This was because the pharmacological inhibition of glycolysis with 2-Deoxy-D-glucose (2-DG) did not affect the differentiation of osteoclasts and viability but effectively blocked the resorption of bone under both in vitro and in vivo conditions [18]. In consistent with this, a study demonstrated that the energy required for the differentiation of osteoclasts is mainly derived from the OXPHOS; however, bone matrix degradation by osteoclasts is supported by the glycolytic process [19]. In response to RANKL, osteoclasts showed a marked enhancement in the consumption of glucose and oxygen along with more production of lactate, suggesting the pivotal role of glycolysis and OXPHOS during osteoclastogenesis. In support of this, Li et al. disclosed that along with lactate production, OXPHOS is crucial for osteoclast differentiation [20]. It has been observed that cell proliferation and differentiation differ with the varied concentration of glucose where maximum growth was observed at 20 mM, and differentiation was observed at 5 mM of glucose concentration. These results suggest that osteoclastogenesis may be controlled by alterations in the concentrations of metabolic substrates. To further assess the impact of the metabolic shift on osteoclast differentiation, they treated bone marrow-derived macrophages (BMDMs) in the presence of pyruvate, further inducing mitochondrial respiration. Pyruvate treatment dramatically increased osteoclastogenesis through the extracellular signal-regulated kinase (ERK) and c-Jun N-terminal kinase (JNK) signaling pathways. As a result, it is possible that the metabolism of glucose would accelerate and shift towards mitochondrial oxidation which may boost ATP production and thus osteoclast differentiation [21]. More recently, the potential role of the lactate dehydrogenase (LDH: enzyme converting pyruvate to lactate) has been investigated and it has been observed that during osteoclast differentiation the uptake of glucose and the production of lactate were enhanced in the culture media [22]. By activating LDH, osteoclasts tend to create lactate in the presence of RANKL, which further drives the formation of osteoclasts by encouraging the fusion of osteoclast precursors and by activating NFATc1 signaling [22]. Furthermore, it has been found that a common sign of aging (mitochondrial dysfunction) is linked to decreased osteogenesis and hastened bone loss (Figure 3). These studies suggest that the glycolytic pathway is important for the osteoclast’s differentiation, and enhanced glycolysis further fuels the resorption of bone [17], thereby highlighting the fact that the inhibition of glycolysis ameliorates bone loss in the pre-clinical model of osteoporosis. Altogether, these reports suggest that glycolysis is the hallmark of functionally active osteoclasts and thus offers a novel opportunity for therapeutic targeting of metabolic diseases associated with osteoclasts dysfunction [23].

Glucose is well utilized by the osteoblast and osteoclast cells to augment their differentiation and functional activity. It is interesting to note that osteocytes further oversee the functions of both osteoclasts and osteoblasts in bone tissue. Osteocytes are osteoblasts lineage cells embedded in the bone matrix [4]. It has been observed that the lacuna-canalicular network of osteocytes in the bone matrix is exposed to a hypoxic environment, thus, leading to a different bioenergetic environment in comparison to the other bone cells [24]. Apart from the morphological changes, osteocytes differ from osteoblasts with varied gene expressions. Genes involved in osteoblasts function like collagen 1 type A1 (COL1A1), and alkaline phosphatase (ALP) are downregulated, whereas dentin matrix protein 1 (DMP1) and sclerostin (SOST) are upregulated in osteocytes. SOST acts as a negative regulator of osteoblasts that suppresses bone formation via negatively regulating the Wnt signaling pathway [25]. Moreover, to control mineralization, osteocytes regulate local and systemic phosphate and calcium metabolism. During lactation, osteocytes release calcium from the bones via osteocyte mediated osteolysis in a parathyroid hormone 1 receptor (PTHR1) dependent manner [26]. Based on the hypoxic environment, we can speculate that the metabolism of osteocytes is predominantly dependent on glucose metabolism and glycolysis. Under hypoxic conditions, molecular pathways responsible for shifting metabolism to glycolysis and autophagy pathways are found to be enhanced, whereas mitochondrial metabolism to generate ATP is reduced. Nevertheless, the isoform of Glut that is responsible for the glucose uptake in osteocytes has not been studied in detail. 

### 2.2. Lipid Metabolism in Bone Cells

Most research to date has concentrated on the effect of glucose utilization by bone cells. Nevertheless, the oxidation of fatty acids results in higher energy production and recently researchers have begun exploring the role of lipid metabolism in skeleton homeostasis. Lipids comprise various species including cholesterol, fatty acids, phospholipids, and triglycerides (TGs). In the human bone marrow, less than 3% of phospholipids and 28–84% of neutral lipids, such as TGs, free fatty acids, and cholesterol are present [27]. In the host, lipids are processed via lipid metabolic pathways and supply lipids to bone cells. They are produced both via dietary lipid consumption and de novo lipid biosynthesis. Intestinal epithelium cells digest the dietary lipids and package TGs and cholesterol into chylomicrons. These complexes pass through the lymphatic system and then reach the bloodstream, where they undergo the exogenous lipoprotein pathway and pick up various apolipoproteins (Apo), such as ApoB, ApoC-II, ApoC-III, and ApoE. Because of their insoluble nature, TGs and cholesterol are transported in a complex with other proteins. Lipoprotein lipase (LPL) hydrolyses the TGs carried by chylomicrons in muscle and adipose tissue, releasing free fatty acids for cellular absorption and chylomicron remnants that are later absorbed by the liver. Bone cells are known to express receptors for the uptake of fatty acids and genes responsible for carrying out their metabolism. The majority of ATP is produced by the β-oxidation of fatty acids within mitochondria [28]. Recent advancements suggest the vital role of adipokines (signaling molecules produced by adipose tissues) in regulating bone metabolism. The newly discovered C1qTNF-related protein 4 (Ctrp4) adipokine, belonging to the CTRP family showed robust osteogenic potential. It was shown that a reduction in the levels of Ctrp4 adipokine suppresses osteoblast differentiation in an ovariectomized mice model [29]. Thereby suggesting Ctrp4 as a potent target for the treatment of bone metabolic diseases. 

Research has long been focused on how osteoblasts use glucose, but lipids and their derivatives are now understood to be an essential energy source for osteoblasts. For the uptake and use of circulating lipids, osteoblasts have the required receptors and catabolic enzymes [30]. The importance of lipid metabolism in osteoblasts is further supported by the fact that dysregulation of lipid metabolism may impart a negative impact on the bone-forming functions of osteoblasts. The study reported by Kim et al. showed that osteoblasts require an extracellular lipid source, and it has been observed that stored lipids are not found to be the energy source for osteoblasts’ function [31]. For the uptake of lipids, osteoblasts express low-density lipoprotein receptors (LDLR) and low-density lipoprotein receptors related to protein-1 (LRP-1), scavenger receptor class B type 1 (Scarb-1), (high-density lipoprotein-HDL receptor) where ablation of these receptors impairs the expression of genes necessary for osteoblast differentiation. In addition, osteoblasts also express receptors for the uptake of free fatty acids such as CD36 (transmembrane glycoprotein receptor) where the deletion of CD36 in null mice lowers the bone mass by impairing the formation of bone mass by osteoblasts [32] (Figure 2). Surprisingly, various studies also suggested the potential role of skeleton in lipid homeostasis, where reduced efficacy of osteoblasts and osteocytes in their lipid uptake disrupts whole-body lipid homeostasis. It has been observed that loss of carnitine palmitoyltransferase-2 (cpt2-enzyme involved in mitochondria β-oxidation) impairs the osteoblasts ability to catabolize fatty acids, thereby impairing the whole body’s lipid metabolism [31]. Thus, these studies indicate that fatty acid catabolism is important to provide energy to induce normal bone formation and it would be expected that osteo-anabolic signals should increase β-oxidation. In support of this, a study demonstrated that PTH and 1, 25-dihydroxyvitamin D3 promote the oxidation of palmitate in the osteoblast calvarial cultures and thus enhance osteoblastogenesis [33]. In several bone-loss-related disorders such as aging, anorexia nervosa, and post-menopausal conditions, reduced bone production is correlated with increased bone marrow fat. It has been observed that conditioned medium (CM) collected from the adipocytes drastically reduced the bone morphogenetic protein-2 (BMP-2) induced osteoblast differentiation in mouse bone marrow cells (BMCs) [34]. Accumulation of lipid droplets in osteoblasts fuels the supply of substrates such as fatty acids and thus supports the osteoblast’s differentiation procedure. Interestingly, it was observed that lipid droplet accumulation seems to be much less at the stromal cell stage (day 0) and committed osteoblast progenitors (day 2), whereas at the mature osteoblasts stage (day 7) it is high. Thus, pointing toward the rapid breakdown of fatty acids at early stages in comparison to the late mature osteoblasts stage. In addition, it has been observed that the protein associated with the lipid droplets is perilipin (PLIN) and the expression of this protein is found to be maximal at the osteoblast’s precursor stage [35]. However, the mechanism responsible for the breakdown of these lipid droplets in osteoblasts needs further investigation. It is also important to ask why the usage of lipids differs at each stage of osteoblast maturation. In addition, it is important to investigate the inhibitory mechanism that reduces the breakdown of lipid droplets at the mature stage. Moreover, a study suggested that intermediates of the cholesterol biosynthesis pathway are important for the development of mature functionally active osteoblasts from the MSCs. Treatment with the cholesterol biosynthesis pathway inhibitor (Mevastatin) suppressed the mineralization by suppressing the activity of the alkaline phosphatase (ALP) gene [36]. However, treatment with mevastatin did not reduce the expression of osteocalcin. 

Cholesterol is a vital component of the plasma membrane that plays a crucial role in the formation and survival of osteoclasts. A study suggested that cholesterol in the membrane of monocytes is essential for the generation of multinucleated osteoclasts via promoting the fusion of mononuclear cells. In addition, treatment of cells with the hydroxy-3-methylglutaryl coenzyme A (HMG-CoA) reductase inhibitor (simvastatin) suppressed the generation of osteoclasts in a time and dose-dependent manner [37]. Several studies reported that high-density lipoprotein (HDL) induced the efflux of cholesterol from the osteoclasts and promoted the apoptosis of osteoclasts. However, the mechanism underlying the HDL-mediated inhibition of osteoclasts has not been investigated for long. In 2004, a study demonstrated that HDL induces apoptosis of osteoclasts by blocking the mTOR, AKT, and S6K signaling pathways [38]. It is off note that the removal of cholesterol further reduced the activity of V-ATPase in osteoclasts by disturbing the lipid rafts and thus affecting bone resorption [39]. Recently, a study revealed that treatment with HDL (600 ng/mL) reduced the fusion index of osteoclasts, but knockdown of ATP binding cassette subfamily G member 1 (ABCG1) expression suppressed the HDL3-induced efflux of cholesterol and thus inhibited the reduction in the generation of osteoclasts [40]. This study thus suggests that HDL3 induces cholesterol efflux from the osteoclasts by altering the expression of ABCG1 which further induces its apoptosis by impairing the cholesterol biosynthesis in osteoclasts. Unsaturated fatty acids (UFAs) have shown various beneficial effects on the bone. A study reported that osteoclasts and osteoblasts express a free fatty acid receptor-4 (FFAR-4) which aids the uptake of UFAs by osteoblast and osteoclast cells [41]. It has been observed that UFAs suppress osteoclastogenesis and promote osteoblastogenesis via the β-arrestin 2 (βarr2) signaling axis [41]. In consistent with this, recently a study highlighted the bone health modulatory potential of long-chain polyunsaturated fatty acids (LCPUFAs). Treatment with N-3 LCPUFAs inhibited the osteoclasts generation via reducing the OPG/RANKL signaling axis, whereas n-3 polyunsaturated fatty acids (PUFAs) promoted osteoblast differentiation and osteoblast activity by reducing the expression of PPARγ [42]. Together, these studies strongly suggest the pivotal role of lipid metabolism in bone health. However, no study to date has reported the plausible role of lipid metabolism in osteocytes, thereby opening a novel avenue for future research in the field

### 2.3. Glutamine Metabolism in Bone Cells

The most prevalent amino acid in human plasma is glutamine, which typically ranges in concentration from 500 to 700 μM. Apart from directly contributing to protein biosynthesis, glutamine is an essential carbon source and nitrogen group donor for the synthesis of nucleotides, amino acids, glutathione, etc. Recently, studies have revealed the potential role of glutamine metabolism in osteoblastogenesis. In the osteoblast calvarial cultures, glutamine is required for matrix mineralization. It has been found that as BMSCs age, their glutamine intake considerably decreases, which is associated with decreased osteoblast development. Additional isotope tracing has demonstrated that glutamine is converted to citrate via the TCA cycle and helps osteoblasts to produce energy [43]. To understand how osteoblasts uptake and utilize amino acids, a study recently identified alanine, serine, and cysteine transporter 2 (ASCT2 denoted as SLC1A5). A study found that the deletion of SLC1A5, a Na^+^-dependent amino acid exchanger, affects the glutamine and asparagine absorption necessary to maintain the amino acid homeostasis in osteoblasts [44] (Figure 3 and Figure 4). A study reported by the same group highlighted the biphasic modulation of glutamine uptake in response to the Wnt signaling pathway during osteoblast differentiation. By utilizing the cellular biology approaches, another amino acid transporter encoded by the Slc7a7 gene was identified that promotes the uptake of glutamine. Of note, targeting either SLC1A5 or SLC7A7 abrogated the glutamine uptake and thus osteoblast differentiation [45]. Osteoblasts are thought to be continuously produced by skeletal stem cells (SSC) throughout life. Nevertheless, under specific circumstances, the SSC population can be improperly designated or not maintained, leading to a reduction in osteoblast development, a reduction in bone mass, and in extreme cases, osteoporosis. In a variety of illnesses, glutamine metabolism has come to be recognized as a crucial regulator of numerous cellular functions. The rate-limiting initial step in glutamine metabolism is the deamination of glutamine to glutamate by the enzyme glutaminase (GLS). By employing metabolic and genetic approaches, a study demonstrated that glutamine metabolism and GLS activity are required to mediate osteoblast differentiation [46]. Genetic inactivation of GLS1 also abolishes PTH-induced osteoblastogenesis [47]. In contrast to these studies, Gayatri et al. recently reported that glutamine regulates both the cellular sensors mTORC1 and mTORC2 and decides the fate of MSCs. A higher concentration of glutamine by inducing the hyperactivation of mTORC1 suppressed the mTORC2 (known to stabilize Runx2 via inhibiting GSK3β) [48]. Activated GSK3β by causing the ubiquitination of Runx2, suppress the differentiation of MSCs to osteoblasts, thereby *suggesting the importance of glutamine metabolism in osteoprogenitors, i.e., bone formation.*

Recently, a study demonstrated that glutamine metabolite α-ketoglutarate (α-KG) suppressed osteoclastogenesis by upregulating the expression of solute carrier family 7 member 11 (Slc7a11) [49]. Given that, Slc7a11 suppressed the osteoclastogenesis by controlling the RANKL-induced reactive oxygen species (ROS) generation [49]. However, the role of glutamine metabolism in osteoclast differentiation and bone resorption has not been investigated to date. Moreover, the role of glutamine metabolism in modulating osteocyte development and its maintenance has not yet been explored.

### 2.4. Insulin Signaling

A study demonstrated that the deletion of the insulin receptor (InsR-tyrosine kinase) in muscles (major site of glucose uptake) did not affect blood glucose levels, the concentration of insulin, and glucose tolerance; thus, suggesting that other tissues might be involved in the glucose metabolism such as bones. Interestingly, several studies reported that insulin exhibits osteogenic properties under both in vitro and in vivo conditions. Studies highlighted that osteoblasts express the InsR and respond to insulin by increasing cell proliferation, collagen synthesis [50], and enhanced uptake of glucose [51]. In consistent with this, a study reported that knock-out of the InsR in osteoblasts reduced bone formation by decreasing the tissue bone volume. New research suggests the sensitivity of osteoblasts to insulin and their role in maintaining global energy homeostasis. In murine osteoblast cultures, insulin aids the uptake and oxidation of ^14^C glucose in a dose-dependent manner. In undifferentiated osteoblasts, a higher expression of Glut-1, Glut-3, and Glut-4 is observed, whereas during osteoblast differentiation a higher expression of Glut-4 is observed that enhances glucose uptake by up to five-fold. Furthermore, mice lacking the expression of Glut-4 in osteoblasts and osteocytes (ΔGlut4) showed normal bone micro-architecture but possessed enhanced peripheral fat in association with hyperinsulinemia [52]. Surprisingly, these mutant mice showed reduced serum levels of cross-linked C-terminal telopeptide (CTX), which is further indicative of osteoclast activity. In addition, treatment with insulin reversed the skeletal alterations in type 1 diabetes mice and favored the healing of fractures [53]. Nevertheless, the critical role of insulin signaling in osteoclasts has not been explored yet.

### 2.5. Role of Hypoxia in Bone Metabolism

The best possible health of cells and tissues depends on an adequate supply of oxygen. Under hypoxic environments where there is an insufficient supply of oxygen, cells must alter their molecular and physiological processes to extend their survival. Evidence suggests that hypoxia may have an impact on bone health in this situation. Kang et al. reported that osteoblasts specifically activate the hypoxia-inducible factor-1α (HIF-1α) pathway and impair osteoclast differentiation. A study reported that reducing pO2 from 20% to 2% suppressed the mineralization eleven-fold in the rat primary cultures. It also decreased the expression of ALP and osteocalcin, thus suggesting the inhibition of osteoblast differentiation. Importantly, a hypoxic condition induced the reversible state of quiescence in osteoblasts leading to apoptosis [54]. Hypoxia impairs mitochondrial respiration by reducing the expression of genes involved in respiration and the oxygen consumption rate (OCR) along with increasing the activity of the glycolytic pathway (determined by the enhanced lactate production) [55]. Osteoblasts residing in the bone marrow experience hypoxia which, by stabilizing the expression of HIF-1α, impacts the formation of bone [56]. Hypoxia-inducible factor (HIF)-1α affects cellular metabolism by upregulating glycolysis and induces the differentiation of osteoblast precursors into mature osteoblasts [56]. HIF proteins such as HIF-1α or HIF-2α, which are also the transcription factors that mediate hypoxia-related pathways, are often highly expressed in hypoxic circumstances. HIF-1α is constitutively hydroxylated in the presence of oxygen and is then targeted for proteasomal destruction [57]. Under hypoxic conditions, however, this does not happen, allowing HIF- 1α to stay stable in the cell and be able to dimerize with HIF- 1β before binding to the hypoxia-response element and starting the transcription of HIF target genes [57]. Studies suggest contradictory results of hypoxic conditions on osteoblastogenesis; however, the duration and concentration of hypoxic conditions need further investigation. 

Along with modulating osteoblastogenesis, hypoxia is also a crucial inducer of osteoclastogenesis in both murine and human cell culture systems [58]. Previously reported studies showed that low oxygen tension (2%) enhances the differentiation of osteoclasts and bone resorption, whereas hyperoxia reduces osteoclastogenesis [59]. Metabolically active osteoclasts depend on oxidative phosphorylation during differentiation, whereas, for bone resorption, osteoclasts depend on glycolysis. Importantly, HIF activates glycolysis, thus implying that hypoxia exhibits a stronger effect on bone resorption [60]. Several pieces of evidence showed that HIF-1α regulates osteoclastogenesis by controlling the expression of RANKL and OPG in osteoblasts [61]. In contrast to the HIF-1α, HIF-2α directly modulates osteoclast differentiation. It has been observed that overexpression of HIF-2α enhances the number of osteoclasts by inducing the expression of TRAF6 [62]. Together, these studies showed that HIFs play a predominant role in the communication between osteoblast and osteoclast cells. 

Murata et al. highlighted a different mechanism by which hypoxic conditions induce osteoclastogenesis. It has been observed that suppressing the expression of COMMD1 (a negative regulator of osteoclastogenesis) promotes osteoclasts differentiation by inducing glycolysis and thus suggests a novel mechanism by which hypoxia induces bone destruction and inflammation [63] (Figure 4). In human macrophages, COMMD1 suppresses the induction of NF-κB signaling, E2F1-dependent metabolic pathways, and creatine kinase B (CKB) activation induced by the RANKL [63]. Altogether, these studies suggest the requirement of oxygen for bone cell formation and mineralization along with inhibiting osteoclastogenesis.

## 3. Bone Cells and Global Energy Metabolism

The skeleton system is a multitasking tissue that along with performing mechanical, hematopoietic, and metabolic functions also acts as an endocrine organ (ascribed particularly to osteoblasts) [64]. Moreover, studies indicate that the skeleton system controls the whole energy metabolism of the host. Various studies demonstrated that insulin secretion promotes osteoblasts to release an osteocalcin hormone that further enhances whole-body glucose metabolism by acting on the pancreas and other tissues. Osteocalcin exists in two forms: carboxylated and undercarboxylated. A study indicated that a high-fat diet in mice enhanced the uptake of saturated fatty acids by the osteoblasts, which, by promoting the ubiquitin-mediated degradation of insulin receptors, impaired the production of osteocalcin, thereby leading to reduced serum levels of undercarboxylated osteocalcin [65]. Together, these events aggravate insulin resistance in muscle and white adipose tissues (WAT). Osteocalcin is not the only gene affecting glucose metabolism. In addition, Esp is a gene encoding for an intracellular tyrosine phosphatase (OST-PTP) in osteoblasts which regulates metabolic functions differently in comparison to osteocalcin and thus keeps all metabolic functions in check. These study highlight the significance of skeletal cells (endocrine critical network) in controlling global energy metabolism (Figure 5). Along with OCN, osteoblast cells express various other factors such as OPG, bone morphogenetic protein (BMP), osteopontin (OPN), etc. Several emerging pieces of evidence point towards the pivotal contribution of these factors in regulating bone and whole-body energy metabolism. By influencing the release of OCN, a study found that OPG can control bone resorption and glucose metabolism [64]. A study reported that higher levels of OPG along with the soluble RANKL, insulin resistance markers, and C-reactive protein were found to be higher in the prediabetic group in comparison to the healthy group [66]. Moreover, higher levels of OPG were also found to be associated with obese adolescents [67], suggesting towards OPG’s potential to regulate glucose homeostasis under both physiological and metabolic diseases. 

BMSCs can release OPN, which functions as an autocrine cytokine in bone tissues to control bone migration, adhesion, and resorption [68]. OPN-null animals have BMSCs that are more likely to develop into adipocytes and have greater body fat percentages [69]. The release of OPN is primarily correlated with inflammation in other tissues and organs. OPN predominantly heightens inflammatory responses and disrupts glucose homeostasis in adipocytes and hepatocytes, which may further impair phosphatidylcholine and cholesterol metabolism and thus exacerbate nonalcoholic cirrhosis of the liver [64,70]. However, other studies showed that OPN can safeguard β-cells by lowering the production of iNOS and maintaining Ca^2+^ homeostasis [71]. Studies have long shown that energy metabolism is regulated by OPN released by adipocytes, hepatocytes, and macrophages. But it is still unclear whether OPN produced from bone can accomplish the same task. It is plausible to assume that dynamic bone metabolism can alter serum OPN levels because bone tissue is an important source of OPN, and exercise can improve bone metabolism. Therefore, a thorough investigation of the global energy metabolism involved in the process of bone-derived OPN is warranted. 

BMP produced from bone can operate as an autocrine or paracrine factor to control bone metabolism. For instance, in hBMSCs, BMP-2 can enhance the expression of runt-related transcription factor 2 (Runx2) and the inhibitor of differentiation (ID), facilitating the formation of osteoblasts [72]. The metabolism throughout the body is significantly impacted by BMP as well. Adipocyte development, differentiation, maturation, and biological activity can all be regulated by BMPs.

In 2013 Sato et al. reported that the absence of osteocytes leads to lymphopenia and revealed that osteocytes are the important regulators of fat metabolism and lymphopoiesis [73]. It has been observed that in osteocyte-deficient mice, the absorption of inorganic phosphate (pi) increases in the intestine, which further stimulates the pi excretion from the renal system [74]. In addition, a study revealed that by expressing resistin, osteoclasts induce insulin resistance and suggest that suppression of osteoclasts generation could be a potential therapeutic strategy for insulin resistance [75]. Weivoda et al. demonstrated that osteoclast-derived factors Dipeptidyl Peptidase-4 (DPP-4) and glucagon-like peptide (GLP-1) regulate glucose homeostasis and suggested a potential link between bone remodeling and energy metabolism [76]. Altogether these studies suggest the critical role of bone in regulating multiple organs and tissue homeostasis.

## 4. Metabolism and Bone Related Diseases

All these studies suggest the crucial role of metabolism in regulating the differentiation and functional activity of bone cells. Thus, alterations in these metabolic phenotypes result in the development of several bone pathologies such as osteoporosis, RA, etc. In further sections, we exhaustively discuss the crucial role of metabolism in distinct bone pathologies. 

### 4.1. Osteoporosis and Metabolism

Osteoporosis is a metabolic bone disease marked by the accelerated deterioration of bone tissues due to the dysregulated balance of osteoblast and osteoclast cells. Dual-energy X-ray absorptiometry (DEXA) scanning is the current gold standard method to assess BMD. In addition, various circulating biochemical biomarkers are used to monitor bone metabolisms such as CTX-I (bone resorption marker) and PINP (bone formation marker). These markers can be used to monitor a patient’s response to therapies, but they cannot indicate when bone loss or fractures will occur, thus making them ineffective for early diagnosis. As a result, a thorough evaluation of novel biomarkers is required for the early diagnosis and prevention of osteoporosis. A study demonstrated that mitochondrial dysfunction is a potent contributor to osteoporosis where a specific knockout of mitochondrial transcription factor A (TFAM) in osteoclasts increases bone resorption in comparison to wild-type cells [77]. Moreover, osteocyte-specific knockdown of superoxide dismutase-SOD (the enzyme that protects against mitochondria oxidative stress) promotes the development of osteoporosis prematurely. By employing, the omics approach (“genomics and metabolomics”), a study identified a series of metabolites and determined the association of metabolites with bone mineral density variation in osteoporotic patients. It has been observed that urinary excretion of metabolite prolyl-hydroxyproline (a marker of bone collagen degradation) is found to be extensively linked with post-menopausal osteoporosis (PMO) [78]. In addition to being diseases associated with aging, atherosclerosis and osteoporosis have similar pathogenetic processes connected to bone and vascular mineralization. It has been observed that dyslipidemia elevates the concentrations of total and low-density lipoprotein (LDL) concentrations that are further associated with low bone mass and enhanced fracture risk. Dyslipidemia, enhances the activity of the osteoclasts by inducing oxidative stress and systemic inflammation, thereby reducing the formation of bone [79]. 

Lipid metabolism is a vital biochemical process that involves the synthesis and breakdown of lipid species such as cholesterol, fatty acids, phospholipids, and bile acids. It is recognized that lipids have a variety of physiological roles in the body, including the production of phospholipid bilayer membranes, hormone biosynthesis, energy storage, and cell signaling. But in recent years, numerous studies have shown that lipid metabolism may also play a crucial role in inflammatory bone loss disorders. Lipid metabolism is regulated by PPARγ, and it has been observed that the accumulation of lipids by causing its oxidation tends to activate PPARγ. Moreover, activation of PPARγ further promotes the differentiation of bone marrow stromal cells (BMSCs) to adipogenic cells and inhibits its differentiation to osteoblasts. Additionally, several experimental and clinical studies revealed a connection between the production of adipocytes and the decline in BMD. Moreover, using lipid-lowering medications such as statins (used to treat hyperlipidemia) has a positive impact on the management of osteoporosis. Thus, this points to a substantial relationship between bone and lipid metabolism.

To analyze the metabolites in the serum of control and osteoporotic patients untargeted metabolomic and lipidomic approaches via ultra-high performance liquid chromatography and high-resolution mass spectrometry (UHPLC-HRMS) were carried out. Further weighted gene correlation network analysis (WGCNA) data mining indicated that lipid metabolism is dysregulated in osteoporotic patients [80]. Where, glycerol, phospholipids, fatty acids, bile acids, and sphingolipids are majorly affected. Thus, suggesting the alteration of lipid metabolic pathways in bone-related diseases. 

#### 4.1.1. Phospholipids (PL) Metabolism

PL are an integral part of the animal plasma membrane. Enhancements in the levels of two crucial PL such as phosphatidylcholine (PC) and lysophosphatidylcholine (LPC) are suggestive of oxidative stress. In the case of osteoporosis, enhanced plasma levels of LPC and PC are indicative of oxidative stress that may lead to enhanced bone loss and the development of osteoporosis [81]. 

#### 4.1.2. Sphingolipids Metabolism

Sphingolipids include glycosphingolipids (GSL), sphingomyelin (SM), and ceramide. SM regulates cell growth and differentiation whereas GSL plays a crucial role in tissue development and function by regulating the interaction between the cells and matrix. Under the continuous action of acid hydrolase, GSL degrades into ceramide (inhibitor of proliferation and inducer of apoptosis). In the pre-clinical mice model of osteoporosis, SM was found to be reduced, whereas GSL and ceramide were observed to be significantly enhanced [81]. Thus, suggesting dysregulated sphingolipid metabolism in osteoporosis. 

#### 4.1.3. Bile Acid Metabolism

Bile acids are strong detergents that play a crucial role in the processing of lipids and cholesterol. They perform important roles in the gastrointestinal and hepatobiliary systems, are necessary for the absorption of dietary lipids and fat-soluble vitamins, and maintain the equilibrium between the liver’s production of cholesterol and the excretion of cholesterol. In the liver, cholesterol is oxidized primarily to form bile acids. Due to their toxicity to membranes and epithelial cells, bile acid production and localization are largely controlled. Bile acid synthesis occurs by two possible routes as conventional/classical (neutral) approach and the alternative (acidic) pathway. About 90% of bile acid synthesis in the liver occurs via the traditional pathway. The alternate pathway is largely extrahepatic and generates the remaining 10% of bile acids. 

It has been observed that in osteoporosis, bile acids metabolism shifts to an alternative pathway from the classical pathway as investigated by the enhanced ratio of cholic acid (CA): chenodeoxycholic acid (CDCA) [80]. Gut microbiota is the pertinent factor in maintaining bone health. Moreover, it has been observed that dysbiosis of GM alters the bile acids metabolism by altering the production of secondary bile acids as demonstrated according to the ratio of deoxycholic acid (DCA): to CA. CA formation is found to be affected by the bacterial 7 α-dehydroxylase enzymes in the gut which tend to produce more DCA that has a well-known cytotoxic effect and results in damage to the mitochondrial membrane [80]. Thus, this study highlights the importance of altered bile acid metabolism in bone-related diseases, especially osteoporosis. However, a further in-depth mechanistic approach is needed for highlighting the specific role of bile acids in the cellular metabolism of bone cells.

### 4.2. Metabolism in Rheumatoid Arthritis (RA)

RA is a chronic, systemic inflammatory illness associated with synovitis as the predominant pathological alteration. Inflammation observed in RA patients is found to be dependent on distinct immune cell subsets where each immune cell possesses unique metabolic demands. For instance, Tregs use lipids via mitochondrial β-oxidation and the production of ATPs by OXPHOS, but effector T cells depend on glycolytic metabolism for their development and effector roles [82]. While naive B cells are kept in a low metabolic state, their activation depends on metabolic switching toward OXPHOS [83]. Similar to this, when there is inflammation, M1 macrophages (inflammatory) employ glycolysis while M2 macrophages (anti-inflammatory) utilize β-oxidation [84]. High energy requirements are associated with autoinflammatory responses in RA, which also involves increased lipogenesis, glucose and glutamine consumption, and a shift away from OXPHOS toward cellular glycolysis for energy metabolism. For instance, persistent T cell mitochondrial hyperpolarization caused by hypoxia in the RA synovium increases glucose consumption and ATP generation [85]. The immunological microenvironment of RA patients encourages the immune cells and stromal cells to undergo metabolic reprogramming and makes the immune cells dysfunctional. The transformation in cell metabolism from a passive regulatory state to one that is a highly metabolically active state alters the redox-sensitive signaling pathway and causes an accumulation of metabolic intermediates, which can then function as chemical signals and exacerbate the inflammatory response [85]. Thus, this altered metabolism in immune cells leads to the pathogenesis of RA. Furthermore, the reprogramming of glycolysis in RA is significantly influenced by mitochondrial dysfunction. These metabolic alterations could serve as prospective RA treatment targets. Disrupted glycolysis in the synovial fibroblasts (SF) is also found to be linked with the destructive phenotype in RA. Recently, a study demonstrated that the soluble molecules secreted by the T helper cells (Th) induce the SF towards enhanced glycolysis and to an inflammatory state [86]. Thus, modulation of the glycolytic pathway may be a novel treatment approach to reduce the pro-inflammatory characteristic of SF, thereby suggesting glycolysis is a sweet target for the treatment of RA. More recently, a study identified the pathogenic behavior of eukaryotic elongation factor-2 (eEF2K) in RA where the inhibition of eEF2k suppressed the glycolysis in RA [87]. Though several studies highlighted the dysregulation of the glycolytic pathway in RA, no study investigated the alterations in the lipid metabolism in RA and thus needs further investigation.

### 4.3. Metabolism in Osteoarthritis

Osteoarthritis (OA) is a degenerative joint disease marked by high levels of clinical heterogeneity and low-grade inflammation. Although OA was often thought of as a “wear and tear” condition, it is now widely acknowledged that it is a low-grade inflammatory condition that affects the entire joint. A crucial factor involved in the advancement of OA and cartilage degeneration may be the aberrant chondrocyte metabolism, which is in consequence to alterations in the inflammatory milieu [88,89]. When exposed to environmental stress, chondrocytes often switch from one metabolic pathway to another, such as from oxidative phosphorylation to glycolysis, to adapt their metabolism to microenvironmental changes. Normally quiescent articular chondrocytes are triggered by stressors such as proinflammatory cytokines, prostaglandin, and ROS, which cause them to change their phenotype and cause additional disruption of the cartilage’s homeostasis and metabolism. This phenomenon is now known as “chondrosenescence” [90]. Alterations in metabolic processes linked to switching between these pathways include mitochondrial dysfunction, increased anaerobic glycolysis, and altered lipid and amino acid metabolism. The AMP-activated protein kinase (AMPK) and mammalian target of rapamycin (mTOR) pathways is primarily responsible for controlling the transition between oxidative phosphorylation and glycolysis. Thus, potential targets for future therapeutic approaches may include metabolic alterations in chondrocytes and other synovial joint cells.

### 4.4. Metabolism in Spondylitis

Ankylosing spondylitis (AS) is a condition that primarily affects the spine and sacroiliac joints and is categorized as spondyloarthropathy. AS typically manifests as chronic back pain. Males are more likely to have AS, which often starts in the third or fourth decade of life and manifests as inflammatory back pain. By the time the condition has progressed to an advanced stage, it may have already caused spinal deformity, mobility restrictions, and a correspondingly reduced quality of life. But the cause of AS is still unknown. A recent study sought to identify trustworthy serum biomarkers for the detection of AS and explore the systemic metabolic changes linked to AS via a metabolomics approach. Thorough investigation revealed that AS patients exhibit altered choline metabolism, lipid metabolism, carbohydrate metabolism, glutamine metabolism, etc. [91]. Moreover, the chances of developing diabetes mellitus are found to be higher in AS patients in comparison to the control group. In consistence with this, more recently, a study reported an altered fatty acid metabolism in AS patients. Metabolomic analysis via time of flight (TOF) mass spectrometry revealed enhanced levels of phosphatidylcholine (PC) and palmitic acid (PA) in the plasma of AS patients and suggested that modulation of diet accordingly with PC and PA could be a potent therapeutic target [92].

### 4.5. Metabolism in Periodontitis

Periodontitis is the most common inflammatory oral disease that, in its severe form, leads to bone loss, which is required to support the teeth for mastication. In a healthy state, the oral microbiome is home to hundreds of different bacterial species that coexist in homeostasis with the host. Under certain circumstances, disturbance in the ecological balance leads to the dysregulated association between the host response and microbiota. Various metabolic products affect the connection between the host and bacteria, which further promote pathogenesis. One of the important metabolites is succinate (crucial TCA cycle intermediate metabolite) and its accumulation in a gingival crevicular fluid is linked with the progression and pathogenesis of periodontitis. In 2017, Guo et al. reported that succinate promotes the differentiation of osteoclasts [93]. Chouchani et al. highlighted that accumulation of succinate occurs when there is an imbalance between the energy demand and the oxygen supply [94]. Succinate builds up first in the mitochondria, then the cytosol, and finally is released into the extracellular space, where it activates SUCNR1 on the target cells in either autonomous or non-autonomous modes. Activation of SUCNR1 by succinate is linked to several illnesses, including periodontitis. Recently, a study reported that in severe periodontitis, the elevation of succinate is linked to the extent of periodontitis by enhancing the periodontal pathogens, inducing inflammation, and eventually bone resorption [95]. Moreover, treatment with the SUCNR1 antagonist reduced the inflammation and thus prevented the bone loss induced by the inoculation of periodontal pathogens. 

### 4.6. Metabolism in Osteogenesis Imperfecta

The most prevalent heritable bone fragility disorder is osteogenesis imperfecta (OI) which is typically linked with autosomal dominant mutations in the COL1A1 or COL1A2 type I collagen alpha chain genes. Diaz et al. reported an altered whole energy metabolism in the OI mice model [96] and suggested that OI is a metabolic disease. Recently, a study reported that *oim/oim* mice showed mitochondrial dysfunction in the skeletal disease, as evidenced by the altered fatty acid oxidation, ROS generation, etc. [97]. However, further research is extensively required for dissecting the association between the different type I collagen mutations and the mitochondrial/cytosol respiration in both the OI mice models and patients. 

### 4.7. Metabolism in Diabetes Associated Bone Diseases

Under the hyperglycemic condition, activation of PPARγ induces adipogenesis and enhanced bone loss. It has been observed that poor control of glucose levels in diabetic patients suppresses the Runx2 involved in osteoblastogenesis [98]. In addition, enhanced adipogenesis further augments bone loss. In post-menopausal women with type 2 diabetes mellitus (T2DM) showed an inverse correlation between the adipose tissue and BMD. Several studies revealed that adipose tissue (mainly visceral fat) synthesizes adipokines such as omentin-1, adiponectin, and visfatin, which negatively impacts bone health, whereas leptin exerts positive effects [99]. Recently, the role of adipo-myokine such as irisin has been described, which induces the uptake of glucose by muscle cells. In addition, irisin promotes osteoblasts differentiation and reduces the number of osteoclasts. In diabetes, irisin resistance has been observed with a resulting loss of all the bone-beneficial effects of irisin. Hyperglycemia leads to the production of advanced glycation end products (AGEs) via non-enzymatic pathways and AGEs have a well-established detrimental effect on bone health by affecting the extracellular matrix (ECM) and the vessels. In addition, AGEs, reduce the formation of bone by inducing the synthesis of sclerostin (inhibitor of osteoblasts) by osteocytes. Recently, a study reported that under high glucose or diabetic conditions glutamine causes the activation of mTORC1 that, by suppressing the mTORC2, leads to the inactivation of Runx2. It has been observed that metformin (anti-diabetic drug) rescues the Runx2 by inhibiting mTORC1. This occurs when the mTORC2/AKT-473 axis is modulated and thus may inhibit diabetes-induced adipogenesis and reduce bone mass [48]. 

### 4.8. Obesity-Related Bone Loss and Metabolism

Obesity is a serious health concern among children affecting various organs of the body including bone tissue. A study demonstrated that mice fed with a high-fat diet (HFD) for 3 weeks showed a lower bone mass and an enhanced number of osteoclast cells. Moreover, bone marrow cells showed higher frequencies of osteoclast precursors in the BM [100]. In addition, HFD along with increasing the osteoclast precursors also enhances the bone resorptive activity in osteoclasts determined by the augmented levels of osteoclastogenic factors such as the tumor necrosis factor (TNF), RANKL, and PPARγ. Strikingly, osteoblast and adipocyte differentiating factors are also enhanced in the bone marrow, thus suggesting that HFD augments bone loss predominantly by potentiating osteoclasts differentiation and their bone-resorbing activity [100]. 

## 5. Gut Microbiota & Bone Metabolism

Despite the fact that a number of presently marketed medications reduce osteoclast differentiation or activity, there is still an opportunity for advancement, notably in the areas of patient side effects and specific targeting. Over the past two decades, it has become clear that gut microbiota plays a critical role in the maintenance of bone health as well as in the etiology of disease. Gut-associated metabolites (GAMs) are the small molecules that are being produced as intermediate or end products of microbial metabolism. These are one of the keyways through which gut microbiota interacts with the host. These by-products may result from bacterial metabolism of food substrates, alteration of host molecules such as bile acids, or production by bacteria themselves. Signals from these metabolites affect the host’s energy metabolism and immune homeostasis. Numerous studies on bone-related diseases have described modifications in the microbiota’s composition and function. Additionally, several groups of metabolites have been linked to the pathophysiology of bone-related diseases, including bile acids, short-chain fatty acids (SCFAs), and indole derivatives (tryptophan metabolites) (Table 1). The effects of microbial by-products on the immune system of the host are well known. The main metabolites derived from microbial fermentation of dietary fibers in the intestine are SCFAs including acetate (C2), propionate (C3), butyrate (C4), pentanoate (C5), and hexanoate (C6) that affect the local and systemic immune functions. Lucas et al. reported that SCFAs, mainly C3 and C4, suppressed the osteoclastogenesis at the early stage by metabolically reprogramming the osteoclasts and thus prevented bone loss in the osteoporotic mice model [101]. It has been reported that treatment with these SCFAs induce glycolysis in precursors of osteoclasts which led to the generation of cellular stress and thus prevented the differentiation of osteoclasts. A study reported that SCFAs exert their effect by binding to FFAR2 receptors, which are primarily expressed by the bone cells, thus suggesting the strong potential of SCFAs to regulate bone health [102]. However, the role of C5 and C6 in modulating the bone remodeling process has not been investigated to date. In addition, medium-chain fatty acids (MCFAs) mainly capric acid are found to suppress the differentiation of osteoclastogenesis. Supplementation of MCFAs also prevented the pathogenesis of osteoarthritis without causing any serious cartilaginous side effects [103]. One of the MCFA, capric acid suppressed the RANKL and lipopolysaccharides (LPS) induced osteoclastogenesis via inhibiting the nuclear factor kappa B (NFκB) and signal transducer and activator of transcription (STAT3) signaling pathways, respectively [104,105]. The farnesoid X receptor (FXR) functions as a bile acid sensor that controls bile acid homeostasis. Cho et al. reported that mouse calvaria and bone marrow cells express FXR, and its expression was found to be enhanced during osteoblast differentiation [106]. In FXR-/- deficient mice, reduction in the BMD was observed along with enhanced TRAP staining at the trabecular area in comparison to the wild-type mice. In addition, it has been observed that BA (chenodeoxycholic acid-CDCA) enhanced osteoblastogenesis by causing the activation of FXR via activating Runx2-mediated ERK and β-catenin signaling pathways. BA supplementation also suppressed osteoclastogenesis [106]. These studies thus suggest the role of bile acids, mainly CDCA, in inducing osteoblastogenesis and suppressing osteoclastogenesis. In support of this, a study recently reported that bile acid is found to be positively associated with BMD and negatively associated with the bone resorption markers such as CTX in post-menopausal osteoporosis [107]. Together these studies suggest the potential of GAMs in inhibiting bone deterioration by altering the metabolism. 

## 6. Therapeutic Interventions on Osteometabolism

Currently, various therapies are being employed for the treatment of bone loss diseases. In the further sections, we highlight the potency of these therapies to target inflammatory bone loss disease by altering the host bioenergetics and metabolism.

### 6.1. Anti-Resorptive Drugs

The cornerstone of treating osteoporosis is antiresorptive therapy. Currently, there are five distinct groups of pure antiresorptive drugs: estrogen, bisphosphonates, denosumab (anti-RANKL monoclonal antibody), selective estrogen receptor modulators (SERMs), and calcitonin. Randomized clinical trials (RCTs) offer solid proof of the effectiveness of antiresorptive treatments in reducing the fractures in the case of postmenopausal osteoporotic women. As bisphosphonates are the first choice for the treatment of post-menopausal osteoporosis and Paget disease. Among bisphosphonates, non-amino bisphosphonates such as clodronic acid are intracellularly converted into the toxic analogue of ATP that induces the apoptosis of cells. On the other hand, amino-bisphosphonates including zoledronic acid (ZA) suppressed the enzyme farnesyl-diphosphonate synthase of the mevalonate pathway. This pathway controls glucose and cholesterol homeostasis. In a retrospective cohort study, it has been observed that intravenous infusion of ZA (5 mg) reduces the atherogenic lipids and glucose in patients suffering from metabolic bone diseases such as osteoporotic and Paget diseases, whereas clodronic acid (1500 mg) failed to do so. This study thus suggests that ZA exhibits the potential to reduce lipid deposition in the atherosclerotic plaque and prevent vascular calcification [121]. Since monocytes and macrophages are the osteoclast precursors, under osteoclastogenic conditions differentiate into mature osteoclasts. A study reported that bisphosphonates (also an immunomodulator) suppress the mevalonate pathway in tissue-resident macrophages, along with preventing protein prenylation, enhancing the immune response to bacterial infections. In 2020, Weivoda et al. reported that post-menopausal women treated with the denosumab showed a reduction in the expression of coupling factors such as DPP4 and GLP-1 that, by controlling the glucose homeostasis, reduced the HbA1C levels in type 2 diabetic patients [76]. Bisphosphonates and other anti-resorptive therapies do exhibit strong anti-osteoclastogenic potential; however, the potency of these therapies in modulating osteoclast metabolism and alleviating bone loss has not been discussed elsewhere, thus opening future avenues in the field.

### 6.2. Osteo-Anabolic Drugs

Osteo-anabolic drugs enhance the formation of bone by activating osteoblast and bone remodeling processes. Teriparatide (recombinant peptide related to amino acids at 1–34 position of human PTH), Abaloparatide, and Romosozumab are the only FDA-approved drugs for the treatment of bone-related pathologies mainly osteoporosis. Osteoporosis is a major complication associated with diabetes mellitus (DM) and diabetes-induced osteoporosis has different pathologies in comparison to post-menopausal osteoporosis. Unfortunately, no specific treatment guidelines are available for the treatment of diabetes-induced osteoporosis. To address this issue, Nomura et al. evaluated the potency of teriparatide, risedronate (bisphosphonate), and calcitonin to enhance bone health in the type 2 DM model. It has been observed that teriparatide and risedronate treatment enhanced the bone micro-architecture along with increasing the BMD in type 2 DM mice [122]. Moreover, teriparatide treatment also reduced the lipid droplets and triglycerides in the fatty liver suggesting towards efficacy of teriparatide to modulate host metabolism in maintaining bone health. Teriparatide showed an acute impact on glucose metabolism; however, when teriparatide is employed for a long time, it subsides the chronic impact on glucose metabolism in osteoporotic patients [123]. Teriparatide is an effective drug that stimulates bone formation by enhancing glycolysis and suppressing the entry of glucose-derived metabolites into the TCA cycle. Mechanistically, teriparatide induces glycolysis via insulin-like growth factor (IGF) and contributes to bone formation [124]. These studies suggest the pertinent role of osteo-anabolic drugs on bone cells via modulating the metabolism in the surrounding and within the bone cells and thus modulating the development and differentiation of bone cells, but the in-depth metabolome of patients in response to these therapies has not yet been determined. 

### 6.3. Steroidal Therapies

Over the past 20 years, there has been a surge in interest in identifying the role of tissue-specific metabolism of endogenous glucocorticoids (GCs) in the aetiology of human disease [125]. In recent years, an explosion in research revealed that excess glucocorticoids adversely affect bone health. Glucocorticoid-induced osteoporosis (GIOP) is a common form of osteoporosis that may enhance the fracture risk in 30 to 50% of patients [126]. In arthritis patients, glucocorticoid application may further enhance the risk of bone and joint damage. In contrast to the effect of the exogenous administration of glucocorticoids, endogenous glucocorticoid supports cartilage growth [127]. Growing evidence points to the importance of localized pre-receptor metabolism of GCs (endogenous and therapeutic) via 11β-hydroxysteroid dehydrogenase (11β-HSD) enzymes (interconvert endogenous GCs between their inactive and active forms) plays a vital role in mediating both their beneficial and detrimental effects on bone homeostasis. Multiple facets of glucose homeostasis are regulated by steroid hormones such as glucocorticoids [128]. 

### 6.4. Vitamin D 

Calcitriol, also known as 1,25(OH)2D3 (1,25D3), is the physiologically active form of vitamin D. This is formed via a multi-step process that begins in the skin and involves the liver and kidney. In response to UV-B exposure, 7-dehydrocholesterol (pro-vitamin D3) is converted into (pre) vitamin D3 (cholecalciferol) in the skin. In the liver and kidney, further hydroxylation at the C25 and 1 position, respectively, results in 1,25 D3. Vitamin D is frequently used in conjunction with other osteoporosis medications to ensure a favorable calcium balance in bone illnesses. Vitamin D is utilized as an anti-rickets medicine that enhances bone mineralization [129]. Thus, the three primary processes in the metabolism of vitamin D are 25-hydroxylation, 1 α -hydroxylation, and 24-hydroxylation, all of which are carried out by cytochrome P450 mixed-function oxidases (CYPs). Emerging evidence indicates that vitamin D controls the activity of bone cells (osteoblast and osteoclast) [129]. Circulating levels of an active form of vitamin D, i.e., 1α,25-dihydroxy vitamin D3 is derived from the renal mediated conversion of 25-hydroxyvitamin D with the help of an enzyme 25D-1α-hydroxylase. Moreover, it has been observed that circulating levels of 25 D rather than 1, 25 D are a good indicator of vitamin D status. Atkins et al. reported that osteoblasts can also metabolize 25 D and secrete 1, 25 D by expressing the cytochrome P450 family 27 subfamily B member 1 (CYP27B1) [130]. Osteoblast cells also upregulate the expression of genes such as OPN OCN, and RANKL [130]. Along with osteoblasts, osteoclasts also express vitamin D receptor (VDR) which in response to vitamin D upregulates the expression of CYP27B1 that further downregulates the expression of genes responsible for osteoclast fusion [131]. Together, these studies suggest that both autocrine and paracrine pathways involved in the vitamin D3 metabolism control the functions of osteoblasts and osteoclasts irrespective of circulating kidney-produced vitamin D levels.

## 7. Conclusions

In this present review, the metabolic characteristics of bone cells are discussed, along with the processes that regulate the energy utilization and the bioenergetics of bone cells. The major metabolic pathways in the bone cells and the activity of important metabolic checkpoint regulators have been uncovered in the article. Targeting metabolism is a successful therapy for several diseases including cancer, thus suggesting that metabolic pathways and networks of bone cells would result in the culmination of innovative therapeutic approaches for pathological bone loss in various inflammatory bone pathologies including osteoporosis, RA, OA, etc. Bone pathologies are categorized as immunological chronic diseases where the role of various immune cells has been well explored such as in the case of osteoporosis (i.e., Immunoporosis) [4,132]. All inflammatory bone loss conditions are classified as metabolic diseases that result either from the alterations in the metabolism of immune cells (Immunometabolism) or from the alterations in the metabolism of bone cells (proposed here as “Osteometabolism”). Thus, understanding the nexus between metabolism and bone health would open up novel avenues for further research and treatment of various inflammatory bone loss diseases (Figure 6).

## Figures and Tables

**Figure 1 cells-11-03943-f001:**
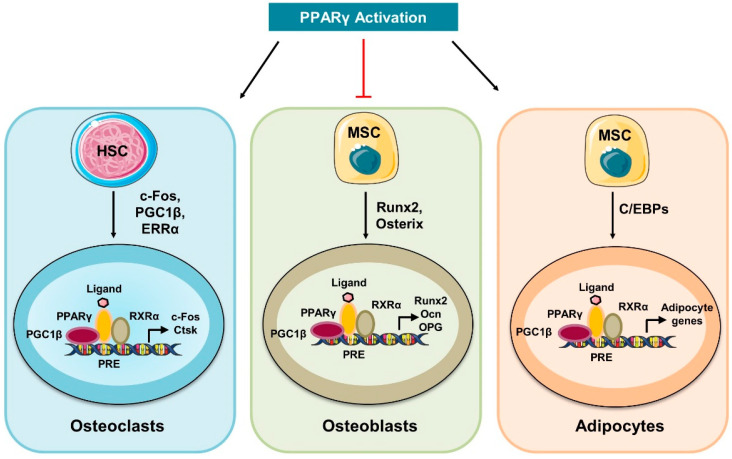
**Activation of PPARγ controls bone cell homeostasis:** Activation of PPARγ promotes the differentiation of HSCs to mature osteoclasts by inducing the expression of osteoclastogenic genes such as cathepsin K (Ctsk) and c-Fos. In response to ligand, activated PPARγ forms a heterodimeric complex with the retinoic X receptor alpha (RXRα) that further recruits the co-activators such as PGC-1β and thus controls the expression of osteoclastogenic genes. In addition, PPARγ inhibits the differentiation of MSCs into osteoblasts and promotes MSCs differentiation into adipocytes.

**Figure 2 cells-11-03943-f002:**
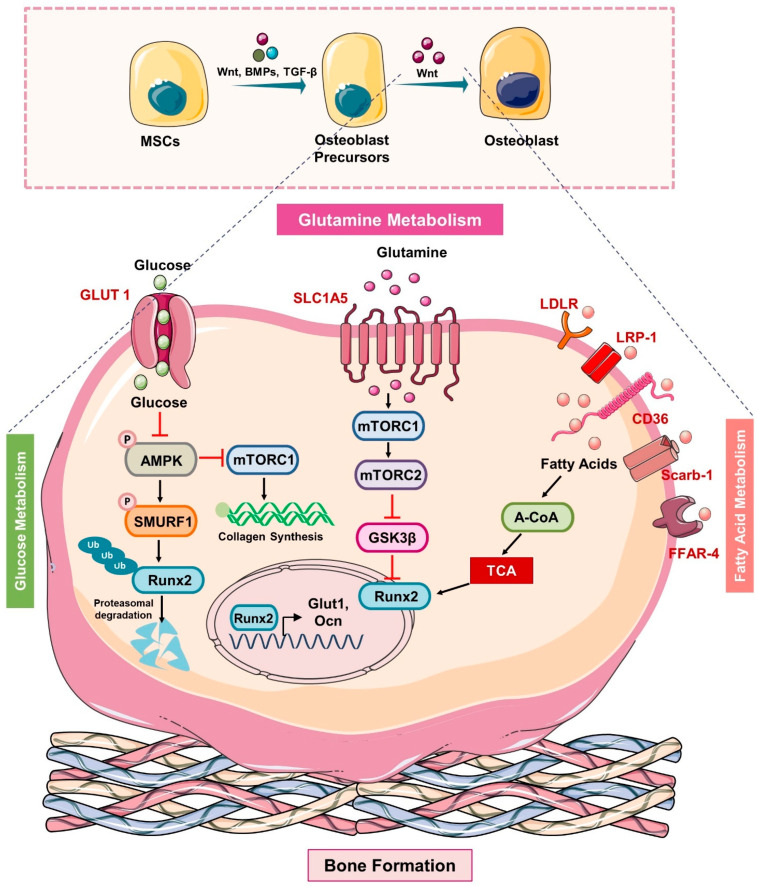
**Metabolic pathways in Osteoblasts:** In response to the Wnt ligand, BMPs and TGFβ, MSCs differentiate into mature osteoblasts. Glut1 controls the uptake of glucose in osteoblast precursors that via inhibiting the phosphorylation of AMP-activated protein kinase (AMPK) suppress the phosphorylation of E3 ubiquitin ligase (SMURF1) that in turn reduces the ubiquitination of Runx2 and thus prevents the proteasomal degradation of Runx2. In addition, glucose uptake also activates the mTORC1 and induces the synthesis of collagen. The uptake of glutamine via the SLC1A5 receptor activates mTORC1 which further inhibits GSK3β by activating the mTORC2 and thus suppresses the degradation of Runx2. In addition, osteoblast precursors express lipid and soluble fat receptors such as the low-density lipoprotein receptor (LDLR), LDL receptor-related protein-1 (LRP-1), CD36, scavenger receptor class B member 1 “Scarb-1”, and the free fatty acid receptor (FFAR-4) that upon ligand binding induces the generation of acetyl-COA that feeds into the TCA cycle to stabilize the expression of osteoblasts genes such as Runx2 and thus enhances the mineralization ability of osteoblasts.

**Figure 3 cells-11-03943-f003:**
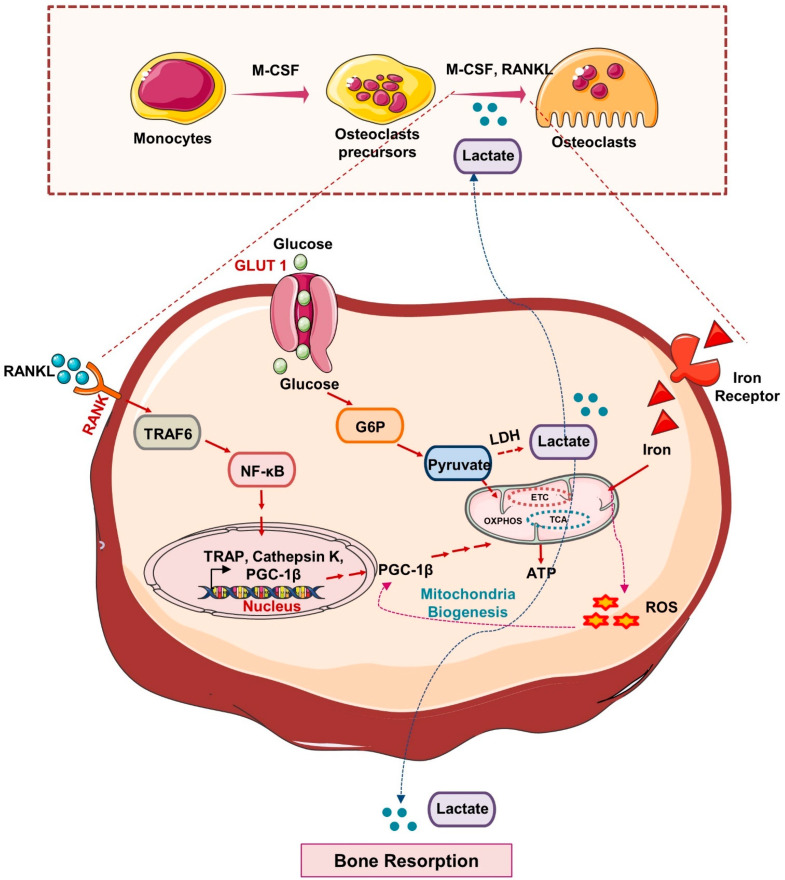
**Metabolic pathways in Osteoclasts:** In response to M-CSF and RANKL, HSCs differentiate into mature osteoclasts. Glut1 controls the uptake of glucose in osteoclast precursors that produce lactate via anaerobic respiration and under aerobic respiration produce pyruvate that undergoes TCA, OXPHOS and promotes ATP generation and thus induces osteoclastogenesis and bone resorption. RANK-RANKL signaling induces the activation of a peroxisome proliferator-activated receptor-gamma coactivator 1β (PGC1β) that, by promoting mitochondrial biogenesis in osteoclasts precursors, further induces its differentiation. In response to iron, the generation of a reactive oxygen species (ROS) stimulates the expression of PGC1β and thus promotes osteoclast differentiation and bone resorption.

**Figure 4 cells-11-03943-f004:**
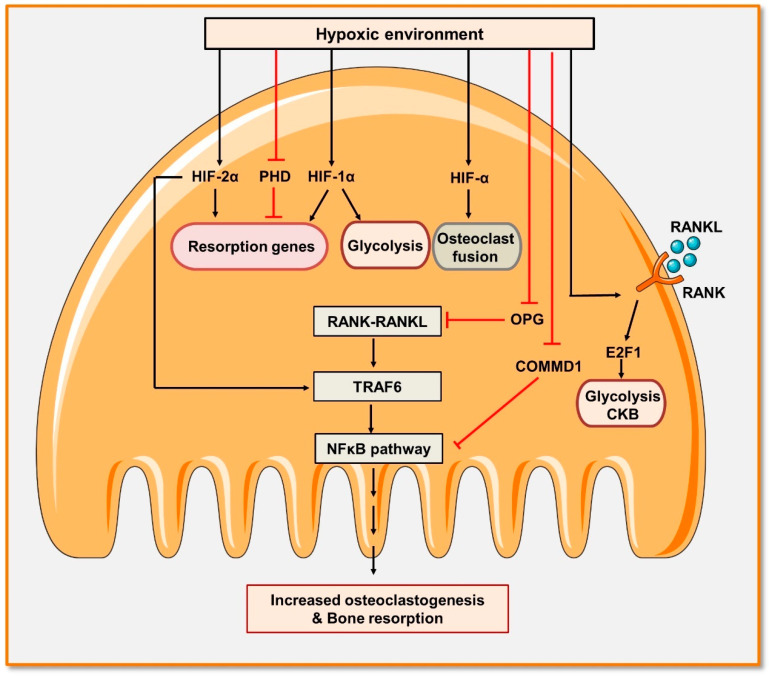
**Hypoxia promotes osteoclast differentiation and bone resorption:** Osteoclastogenesis is accelerated in hypoxic circumstances. Hypoxia-inducible factors 1α and 2α (HIF-1α and HIF-2α) promote the fusion of osteoclasts. HIF-2α also promotes the expression of TRAF6, enhancing the activation of the NF-κB signaling pathway and maturation of osteoclasts. In addition, hypoxia promotes the activation of NF-κB by reducing the expression of osteoprotegerin (OPG). HIF-1α and HIF-2α, as well as the inhibition of the prolyl hydroxylase domain (PHD), stimulate the production of pro-resorptive genes. Additionally, HIF-1 α is important in promoting glycolytic activity either directly or via negatively regulating the expression of the copper metabolism domain containing 1 (COMMD1) gene. Hypoxia also activates RANK-RANKL signaling that, by promoting the expression of elongation transcription factor (E2F), induces glycolysis and thus enhances the differentiation of osteoclasts along with its induced bone resorption activity.

**Figure 5 cells-11-03943-f005:**
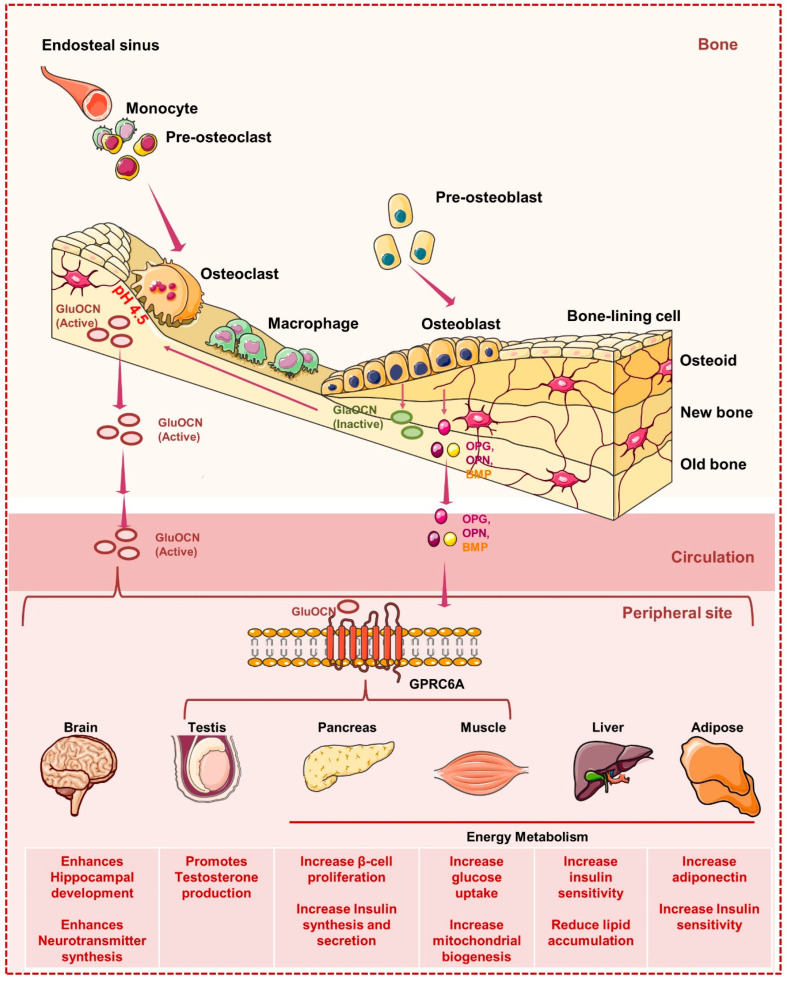
**Bone cells and global energy metabolism:** Osteoblasts release OCN, which is C-carboxylated (GlaOCN), into the bone extracellular matrix (ECM). Osteoclasts form bone resorption lacunae having acidic pH (~4.5) to aid the decarboxylation of GlaOCN to GluOCN. GluOCN enters into the circulation and acts as an endocrine hormone. GluOCN controls global energy metabolism by regulating the uptake of glucose in the muscles, production of insulin in the pancreas, enhancement in the adiponectin expression in the adipose tissues, promoting the proliferation of β-cell in the pancreas. In addition to controlling energy metabolism, OCN regulates the synthesis of testosterone by the Leydig cells and thus controls male fertility. It also facilitates the development of the hippocampus region of the brain. Strikingly, in the tissues like the testis, muscles, and pancreas, OCN regulates its endocrine functions by binding to the G-protein coupled receptor (GPRC6A). However, how OCN acts on the brain, liver, and adipose still need investigation.

**Figure 6 cells-11-03943-f006:**
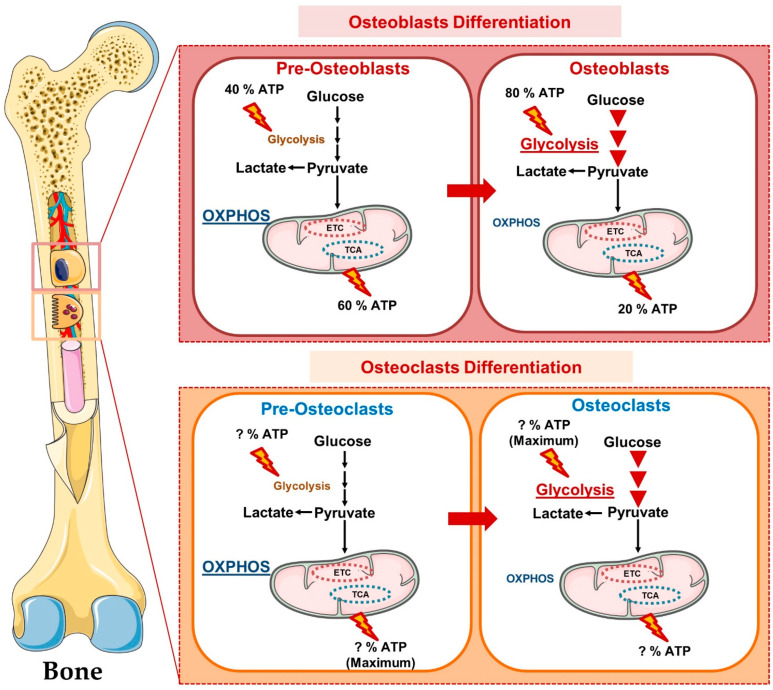
**Osteometabolism (Metabolism of Bone Cells):** In precursors form, osteoclast and osteoblast cells obtain maximum energy from oxidative phosphorylation whereas in mature form these bone cells obtain energy predominantly from glycolysis.

**Table 1 cells-11-03943-t001:** GUT Associated Metabolites (GAMs) and their effect on Bone Cell-metabolism.

S.no.	GAMs	Classification	Effect on Bone Cells	Role in Bone Pathologies	Ref.
1.	SCFAs	Acetate (C2)	Enhance osteoblast differentiation via inducing alkaline phosphatase (ALP) activity and suppress osteoclast differentiation by altering the osteoclast metabolism	Lower levels are associated with osteoporosis and RA progression	[101,108,109]
		Propionate (C3)	Enhance ALP activity in osteoblast and reduce NFAT1 and TRAF6 expression in osteoclast	Lower levels are associated with osteoporosis and RA progression by restoring the GM	[101,108,110]
		Butyrate (C4)	Reduce osteoclast differentiation via downregulating the expression of TRAF6 and NFAT1 along with enhancing the glycolysis	Suppress arthritis via enhancing the aryl hydrocarbon receptor activation in regulatory B cells (Bregs)	[101,111]
		Pentanoate (C5)	Suppress osteoclastogenesis via downregulating the expression of NF-κB in osteoclasts and enhancing the osteoblasts differentiation via increasing the ALP and mineralization potential in osteoblast cells	N.D.	[112]
2.	MCFAs	Caprylic acid (C8:0)	N.D.	N.D.	[113]
		Capric acid (C10:0)	Suppress LPS-induced osteoclastogenesis by inhibiting NO production in a STAT3-dependent manner. In addition, inhibits osteoclastogenesis via suppressing NF-κB signaling	N.D.	[104]
		Lauric acid (C12:0)	N.D.	N.D.	[113]
3.	LCFAs	Myristic acid (C14:0)	Block osteoclastogenesis via suppressing the activation of tyrosine kinases such as Src and Pyk2	N.D.	[114,115]
		Palmitic acid C16:0)	Enhance RANKL induced osteoclastogenesis	N.D.	[116]
		Stearic acid (C18:0)	Enhance RANKL-induced osteoclastogenesis	N.D.	[116]
4.	Primary Bile Acids	Cholic acid (CA)	N.D.	Bile acid positively correlated with the BMD and negatively correlated with the CTX-1 bone turnover marker in osteoporosis	[107,117]
		Chenodeoxycholic acid (CDCA)	Enhance osteoblastogenesis via upregulating the expression of Runx2, ERK, and β-catenin signaling pathways. Also, suppress osteoclast differentiation	N.D.	[106,117]
5.	Secondary Bile Acids	Tauroursodeoxycholic acid (TUDCA)	Increase proliferation and differentiation of osteoblasts	Treatment with TUDCA enhanced the PINP level and lowered the levels of CTX-1 in osteoporotic mice and turn, increase the BMD in osteoporotic mice model	[118]
		Deoxycholic acid (DCA)	N.D.	N.D.	[117]
		Lithocholic acid (LCA)	Decrease the viability of osteoclasts by downregulating the expression of microRNAs such as miR-21a, miR-29b	N.D.	[117]
		Ursodeoxycholic acid (UDCA)	Decrease the viability of osteoclasts by downregulating the expression of microRNAs such as miR-21a, miR-29b	N.D.	[117,119]
6.	Indole Derivatives	Indole-acetic acid (IAA)	N.D.	N.D.	[120]
		Indole-propionic acid (IPA)	N.D.	N.D.	[120]
		Indole-3-aldehyde (IAId)	N.D.	N.D.	[120]
		Indole-3-lactic acid (ILA)	N.D.	N.D.	[120]

N.D: Not determined; SCFAs: Short-chain fatty acids; MCFAs: Medium-chain fatty acids; LCFAs: Long-chain fatty acids.

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
