# Peer review of "Osteometabolism: Metabolic Alterations in Bone Pathologies"

_cells, 2022, doi:10.3390/cells11233943_

Round 1

Reviewer 1 Report

The work titled:

 Osteometabolism: Metabolic alterations in Bone pathologies is undoubtedly a topic of interest for which there have been no updated reviews for at least the last 5 years.  However, the lack of care in referencing the studies mentioned and the repetitive wording of several sentences makes it difficult to read. I recommend major changes before it can be considered for publication. 

 1- Reference 3 The reference should be replaced by a more appropriate one and not only referring to osteoporosis, for example:

“Jann, J.; Gascon, S.; Roux, S.; Faucheux, N. Influence of the TGF-β Superfamily on Osteoclasts/Osteoblasts Balance in Physiological and Pathological Bone Conditions. Int. J. Mol. Sci. 2020, 21, 7597. https://doi.org/10.3390/ijms21207597

2- Check and replace reference 6, I am afraid it is incorrect.

3-Figure 1 is very general, it is desirable to relate it to the metabolism of bone cells with respect to the points mentioned in the corresponding paragraph or to combine it with figure 2, otherwise it is better to eliminate it.

4-The paragraph “These factors have a well-established role in mitochondrial biogenesis, and it has been observed that deficiency…” in page 2, needs to be referenced.

 5-The paragraph “More recently, a study 162 showed that limiting carbohydrates for a short period decreased the levels of circulating 163 bone growth indicators during the rest state.” in page 5, needs to be referenced. Shen, L., Hu, G. & Karner, C.M. Bioenergetic Metabolism In Osteoblast Differentiation. Curr Osteoporos Rep 20, 53–64 (2022). https://doi.org/10.1007/s11914-022-00721-2

6- In section 2.1, “Glucose Metabolism in bone cells”, the first part summarizes glycolysis however this is a general topic that can be synthesized to expand on glycolysis in osteoblasts, there is an excellent review with details on this topic that can be cited, and this reference should be included.

 7-The paragraph “During osteoclastogenesis, expression of Glut-1 and glutamine transporters was observed to be enhanced with a subsequent increase in the genes responsible for glycolysis and glutaminolysis…” in page 6, needs to be referenced.

8-It is incredible that in the paragraph that begins on line 220 and ends on line 239, there is no reference to support the importance of glycolysis in osteocytes. They should certainly be included.

9-Paragraph from line 256 to line 259 mentions lipid proportion in bone marrow, but cortical bone, for example, does not contain them or the amount is minimal?

10-In line 285, “Sarcb-1” should be in parentheses.

11-Line 370, page 11, Slca5 should be SLC1A5? Please homogenize capital letters also.

12-Reference 35 it is correct? It is about Non-Small Cell Lung Cells.

13-In the section 3, “Bone Cells and Global Energy Metabolism”, the role of osteocalcin in energy metabolism, such as insulin secretion, is mentioned. However, the section lacks other important factors secreted by bone that Osteoprotegerin, Osteopontin or Bone morphogenetic protein, should be considered at least briefly. There is an excellent review commenting on these issues, which should be cited. Zhou, R., Guo, Q., Xiao, Y. et al. Endocrine role of bone in the regulation of energy metabolism. Bone Res 9, 25 (2021). https://doi.org/10.1038/s41413-021-00142-4

14-I suggest completing figure 5, with the points cited in my comment number 13.

15- Paragraph from line 496 to 501 needs to be referenced.

16-Paragraph from line 594 to 608 needs to be referenced.

 17-The idea " In a healthy state, the oral microbiome is home to hundreds of different bacterial species that coexist in homeostasis with the host." of line 637 is repeated in the line 647, and the same situation occurs in lines 668 and 671. Authors should be careful in this regard.

 18-Since section 4 deals with bone pathologies and their metabolism and this section includes rheumatoid arthritis closely related to the loss of articular cartilage, why not also include osteoarthritis as the most frequent degenerative joint disease? I consider that it should be included. 

 19-The section titled “Steroidal therapies” again contains no references.

 20-I consider that section number 6 should be placed before the therapeutic interventions.

 21- The graphical abstract should be simplified as it contains too much information.

Author Response

Thanks

Reviewer 2 Report

The present study was about metabolism about osteogenesis and osteoclastogenesis and its metabolic alterations. The topic was very interesting to bone biologist and pathologist but excessive errors should be corrected to published in the present journal.

1. There are a lot of grammatical errors in the abstract and manuscript, and it is very hard to read to author's view. The manuscript should be corrected by professional English editor.

2. Figure 1. is about the general glucose metabolism. However, it disturbs the text flow because of its no relationship with osteometabolism itself. The author should replace to more closed related figure with metabolism about osteogenesis. 

3. The author should explain Pharagraph 2 . line 91 "Upon stimulation with RANKL, enhancement in the number 91 of mitochondria promotes osteoclasts differentiation thus suggesting the higher requirement of energy metabolism" with more detail example and reference. The next line " Furthermore, enhancement in the level of factors responsible for promoting mitochondrial biogenesis in osteoclasts such as peroxisome proliferator activated receptor (PPARγ) and peroxisome proliferator-activated receptor-gamma coac tivator 1β (PGC-1β) was found to be linked with the differentiation and functional activity of osteoclast. " has no evidence and reference in this review. The author should add the detail description to extend the role of PPAR-gamma on bone homeostasis.

4. Pharagraph 2.5 is about the role of hypoxia in bone metabolism, especially osteoblast differentiation. But author should add the detail description about the osteoclast differentiation and activation, actually the effect of hypoxia in bone remodeling. Furthermore, the author already showed the effect of hypoxia on osteoclastogenesis. Author should add the detail description about the osteometabolism in hypoxia such as glycolysis CKB.

5. The author should rearrange the Paragraph 5. That should be about the Therapeutic interventions on osteometabolism. Such as 5.4 Vitamin D need more detail description about the metabolism.

Author Response

Thanks

Round 2

Reviewer 1 Report

The authors made an effort to correct and take into account the suggestions to improve the review, I suggest to accept it for publication.

Reviewer 2 Report

The correction was performed well and it could be publish as current form.